



# Clustering simulated snow profiles to form avalanche forecast regions

Simon Horton[1,2], Florian Herla[1], and Pascal Haegeli[1]

[1]School of Resource & Environmental Management, Simon Fraser University, Burnaby, BC, Canada
[2]Avalanche Canada, Revelstoke, BC, Canada

**Correspondence:** Simon Horton (shorton@avalanche.ca)

**Abstract.** This study presents a statistical clustering method that allows avalanche forecasters to explore patterns in simulated snow profiles. The method uses fuzzy analysis clustering to group small regions into larger forecast regions by considering snow profile characteristics, spatial arrangements, and temporal trends. We developed the method, tuned parameters, and present clustering results using operational snowpack model data and human hazard assessments from the Columbia Mountains of

western Canada during the 2022-23 winter season. The clustering results from simulated snow profiles closely matched actual forecast regions, effectively partitioning areas based on major patterns in avalanche hazard, such as varying danger ratings or avalanche problem types. By leveraging the uncertain predictions of fuzzy analysis clustering, this method can provide avalanche forecasters with a straightforward approach to interpreting complex snowpack model output and identifying regions of uncertainty. We provide practical and technical considerations to help integrate these methods into operational forecasting

practices.

## 1 Introduction

Forecasting avalanche hazard over terrain is fundamental for effectively managing short-term snow avalanche risk (Canadian Avalanche Association, 2016). Forecasters assess the current hazard by interpreting weather, snowpack, and avalanche observations, while also interpreting weather forecasts to predict future hazard conditions. In recent years, forecasters have shown interest in

using numerical snowpack models to reduce their uncertainties (Morin et al., 2020). Models like SNOWPACK (Lehning et al., 1999) and Crocus (Brun et al., 1992) use meteorological data to provide predictions of snow stratigraphy and stability across spatial and temporal scales relevant to avalanche forecasting.

Several recent advancements have considerably enhanced the value of snowpack models for avalanche forecasting. First, improvements to numerical weather prediction models in complex terrain (Lundquist et al., 2020) allow running snowpack

simulations in remote regions (Horton and Haegeli, 2022). Second, new post-processing models establish stronger connections with snow stability (Mayer et al., 2022) and avalanche hazard (Pérez-Guillén et al., 2022). Lastly, applying visual design principles (Horton et al., 2020) and snow profile processing tools (Herla et al., 2021, 2022) can enhance the communication of this information to forecasters. While operational model systems are beginning to incorporate these developments, their adoption into forecasting workflows remains gradual. Therefore, we need to present model output in simple informative ways.





Statistical clustering methods provide an effective means of identifying and summarizing patterns within complex datasets. Bouchayer (2017) was the first to cluster simulated snow profiles by grouping profiles based on the specific surface area of snow layers. Using a dynamic time-warping alignment method developed by Hagenmuller and Pilloix (2016), they constructed a hierarchical clustering tree by comparing vertical sequences of specific surface area. Herla et al. (2021) expanded on this approach by incorporating generic categorical and numeric snowpack properties such as hand hardness and grain type into

the dynamic time-warping process. This enabled them to employ hierarchical clustering methods to group snow profiles based on characteristics relevant to avalanche hazard assessment. Reuter et al. (2023) applied k-means clustering to simulated snow profiles by predicting avalanche problem types from the profiles and then clustering problem prevalences to explore the snow climatologies in the French Alps. While these clustering methods revealed patterns in simulated snowpack properties, they did not fully capture the spatial and temporal patterns important to avalanche forecasters.

To present avalanche forecasters with more accessible and relevant snowpack model information, we developed a method for clustering simulated snow profiles into avalanche forecast regions. This method expands upon the approach introduced by Herla et al. (2021), which partitions snow profiles based on avalanche hazard characteristics, by further addressing the operational requirements for coherent spatial and temporal patterns. We developed the method using operational snowpack simulations and human avalanche hazard assessments from the Columbia Mountains of western Canada. Sect. 2 describes the

study area and data, and then Sect. 3 introduces the clustering method. After selecting appropriate parameters (Sect. 4), we present examples of the clustering results and compare them with human-assessed forecasts in Sect. 5. To help others apply these methods we discuss practical and technical implications in Sect. 6.

## 2  Study area and data

### 2.1  Study area

We developed the clustering method using simulated snow profiles and human-assessed avalanche forecasts in the Columbia Mountains of western Canada (Fig. 1a). The Columbia Mountains have a transitional snow climate prone to storm slab and persistent slab avalanche problems (Shandro and Haegeli, 2018). Variations in weather and snowpack across the range often lead to distinct patterns in avalanche hazard, making it well-suited for exploring spatial clustering methods. For example, storm tracks can impact the northern and southern parts of the range differently, while orographic enhancement often results in

heavier precipitation on the western sides of each subrange.

Public avalanche forecasters at Avalanche Canada, Canada's public avalanche warning service, have divided the Columbia Mountains into 32 permanent subregion polygons. Forecasters aggregate these subregions into larger forecast regions daily based on their assessment of avalanche hazard conditions. In this study, *subregions* refer to the individual subregion polygons and *regions* refer to the aggregated groups of subregions, whether done by human forecasters or clustering methods.





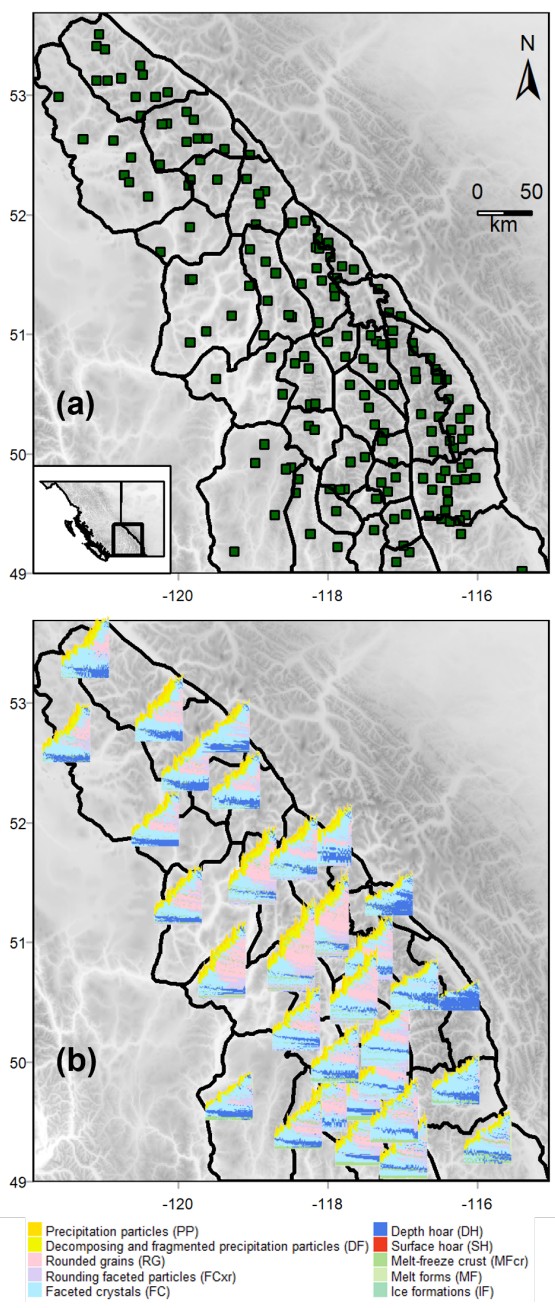

**Figure 1.** Study area and simulated snow profiles are shown with (a) the Columbia Mountains divided into 32 permanent subregion polygons and the locations of individual simulated snow profiles, and (b) the snow profile time series produced by averaging snow profiles within each subregion.



## 2.2 Simulated snow profiles

We obtained simulated snow profiles for the 2022-23 winter season from Avalanche Canada's operational snowpack modelling system (Horton et al., 2023). This model system runs the SNOWPACK model (Lehning et al., 1999) with meteorological data from two numerical weather prediction models. The model generates daily profiles at 168 locations in the Columbia Mountains, representing flat, sheltered terrain at treeline elevations. Since the focus of this paper is presenting a clustering method that applies to any spatially distributed snowpack simulation, the specific methods used for generating these profiles are of limited relevance. Interested readers are referred to Appendix A where the model configuration is explained in detail.

To represent typical treeline elevation snowpack conditions in each subregion, we computed representative profiles using the dynamic time-warping barycenter averaging method developed by Herla et al. (2022). This method aligns profile layers using dynamic time-warping, computes the prevalent grain type mode for each layer, and then averages layer properties of each dominant mode (e.g., thickness, hardness, temperature). Averaging was done for each day of the season to produce 32 snow profile time series representing typical treeline conditions in each subregion (Fig. 1b).

## 2.3 Human-assessed forecast regions

Avalanche Canada issues daily public avalanche forecasts for the Columbia Mountains. Expert forecasters group subregions into semi-homogenous forecast regions and assign danger ratings and avalanche problems to three elevation bands for each region. This study analyzed forecasts between November 26, 2022 and April 24, 2023, starting when daily forecasts were published and ending when the forecasts switched to a single large region for spring conditions. Operational snowpack model data was unavailable for 35 days during this period due to system outages, resulting in 115 days when both model and human data were available for analysis.

## 3 Clustering method

### 3.1 Distance between subregions

Many clustering methods use a distance matrix to quantify differences among data points (Kaufman and Rousseeuw, 2009). A distance metric measures the distance between each pair of points: identical points have a distance of 0, while dissimilar points have larger values. These pairwise distances are arranged in a matrix with rows and columns representing each data point. Our clustering method derives a metric to quantify the distance between subregions in a way that encourages similar subregions to be grouped (Fig. 2). Our distance metric $dist$ considers three relevant criteria:

1. **Snow profile characteristics**: The snow profile distance $dist_{pro}$ quantifies the similarity of snow profiles so that clustering will produce forecast regions with similar avalanche hazard characteristics.

2. **Spatial arrangement**: The spatial distance $dist_{geo}$ quantifies the spatial arrangement of subregion polygons so that clustering will produce spatially contiguous regions.




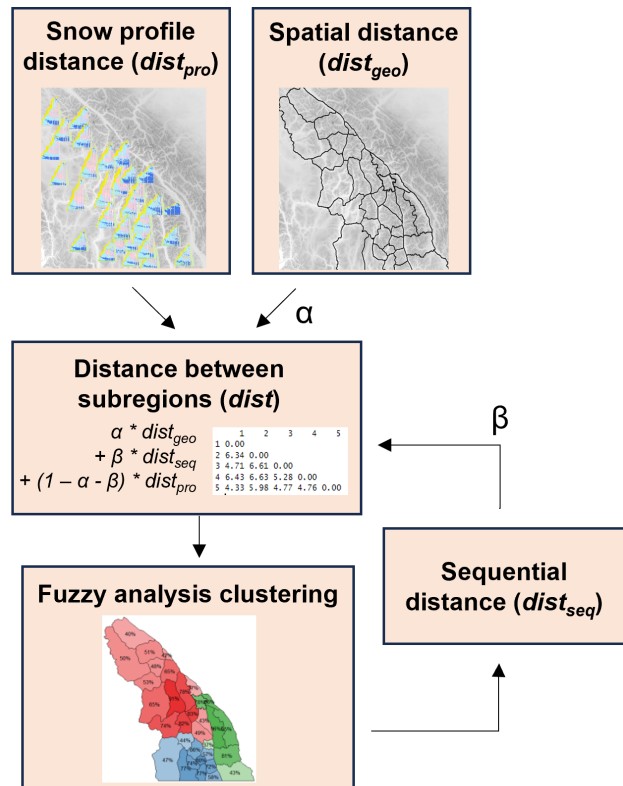

**Figure 2.** The clustering method derives an overall distance matrix integrating three key criteria: snow profile distance, spatial distance, and sequential distance. Snow profile distance is based on snow profile characteristics in simulated snow profiles, spatial distance is based on the arrangement of polygons, and sequential distance is based on the previous day's forecast regions.

3. **Temporal stability**: The sequential distance $dist_{seq}$ quantifies the previous day's clustering results so that clustering will only change forecast regions when there are substantial changes in snow profile characteristics.

After calculating these individual distance metrics, we compute the overall distance between subregions $dist$ using a weighted mean:

$$dist = (\alpha)dist_{geo} + (\beta)dist_{seq} + (1 - \alpha - \beta)dist_{pro} \tag{1}$$

where $\alpha$ is a weight controlling the relative significance of the spatial distance and $\beta$ is a weight controlling the relative significance of the sequential distance.

### 3.1.1 Snow profile distance

We quantify the snow profile distance ($dist_{pro}$) with the snow profile similarity measure introduced by Herla et al. (2021). This method aligns two profiles onto a common height grid using dynamic time-warping then compares the properties of the





layers to assign a similarity score ranging from 0 to 1. The similarity scores are calculated using the *sarp.snowprofile.alignment*
       package for R (Herla et al., 2021, 2022), which offers various approaches to calculate the similarity of aligned profiles. These
       approaches weigh different combinations of grain type, grain size, layer hardness, and instability. We use an approach that
       computes a weighted sum of grain type similarity (37.5 %), hand hardness similarity (12.5 %), and layer stability similarity
       (50 %). We quantify layer stability using the random forest method developed by Mayer et al. (2022) to predict the probability
of instability for each layer in a profile. This approach assigns more weight to unstable layers to reward profiles with similar
       stability patterns. Among the available approaches for quantifying the snow profile similarity in *sarp.snowprofile.alignment*,
       this method most closely aligns with forecasters' criteria for relating snowpack layers to avalanche characteristics. The method
       calculates the pairwise similarity of profiles each day then subtracts them from 1 to produce snow profile distance values.

### 3.1.2    Spatial distance

We consider the spatial distance between subregions to encourage geographically contiguous forecast regions. We designed the
       spatial distance ($dist_{geo}$) to reduce the distance between subregions in close geographic proximity while increasing the distance
       for spatially separated subregions. We derived the spatial distance matrix using a binary neighbourhood-based approach, where
       polygons sharing borders have a distance of 0 and polygons without shared borders have a distance of 1 (Chavent et al., 2018).
       The neighbourhood approach encourages spatially connected forecast regions that are more likely to align with the elongated
shape of mountain ranges than would result from basic Euclidean distances.

### 3.1.3    Sequential distance

When clustering on consecutive days, the arrangement of forecast regions should vary in response to changing avalanche
hazard conditions. However, clustering can be overly sensitive to subtle changes in the dataset which can lead to excessive
changes in forecast region boundaries that may not be practical for forecasting applications. To address this issue, we use a
sequential distance ($dist_{seq}$) to incorporate some weight from the previous day's clustering results in a way that encourages
       subregions to remain in the same groups. Sect. 4.4 explains this approach in detail.

### 3.2    Fuzzy analysis clustering

Given the complexities of avalanche hazard assessment and snow profile data, we chose a fuzzy clustering method to explicitly
highlight the uncertainties associated with assigning data points to clusters. Fuzzy clustering methods use membership degrees
that allow data points to belong to multiple clusters simultaneously (Kaufman and Rousseeuw, 2009).

Our method uses a fuzzy variant of k-medoid clustering called *fuzzy analysis clustering*, or fanny. The fanny method,
implemented in the *cluster* package for R and described by Kaufman and Rousseeuw (2009), assigns each data point $i$
membership values $u_{iv}$ between 0 and 1, quantifying its degree of belonging to cluster $v$. The method aims to minimize
the objective function:



$$\sum_{v=1}^{k} \frac{\sum_{i=1}^{n} \sum_{j=1}^{n} u_{iv}^{r} u_{jv}^{r} dist(i,j)}{2 \sum_{j=1}^{n} u_{jv}^{r}} \tag{2}$$

where $n$ is the number of data points, $k$ is the number of clusters, $dist(i,j)$ is the distance between data points $i$ and $j$, and $r$ is the fuzziness parameter. The fuzziness parameter $r$, whose value can range between 1 and infinity, controls the degree of fuzziness in the clusters. As $r$ approaches 1, clusters become increasingly crisp (i.e., k-medoid clustering), while higher values lead to complete fuzziness (i.e., data points have equal membership in every cluster). The method iteratively defines cluster

centers using the medoid data point and recalculates the membership values until they converge within a specified threshold tolerance.

We arrange the distances between subregions ($dist$) into a matrix and input them into the fanny method to derive cluster membership values for each subregion. This process requires specifying appropriate values for the fuzziness parameter $r$ and the number of clusters $k$, as explained in Section 4.

**4 Optimizing clustering parameters**

To apply our clustering method, several parameters must be defined, including $\alpha$ and $\beta$, which specify how much weight is given to the spatial and sequential distances (Eq. 1), the fuzziness parameter $r$, which determines the crispness of the cluster memberships, and the number of clusters to be estimated $k$ (Eq. 2). Optimal values for these parameters will vary between contexts, so this section outlines methods for appropriate parameter selection.

We used a grid search to systematically explore various parameter combinations (Feurer and Hutter, 2019), then used two approaches to select optimal values from the grid search: cluster validation metrics and comparisons with human-assessed forecast regions. We conducted two grid searches, both using data from the entire study period. The first grid search systematically explored combinations of $\alpha = \{0.05, 0.1, ..., 0.4\}$, $r = \{1.05, 1.10, ..., 1.5\}$, and $k = \{2, ..., 12\}$ with each day treated as independent (i.e., $\beta = 0$). Optimal values from this initial grid search informed a second grid search where sequential clustering was done

over the study period with $\beta = \{0, 0.01, ..., 0.1\}$.

**4.1 Spatial weight**

We examined the spatial arrangement of clusters resulting from the grid search to find the proportion of spatially contiguous versus non-contiguous regions. When considering only snow profile characteristics (i.e., $\alpha = 0$), approximately 42 % of regions were spatially contiguous across all combinations of $r$ and $k$ (Fig. 3). The proportion of contiguous regions increased with

higher values of $\alpha$, exceeding 95 % for $\alpha = 0.3$. The optimal level of spatial contiguity depends on user preferences and the number and arrangement of subregions. While some non-contiguous regions may offer insights into similar snowpack patterns across large distances, an excessive number can result in incoherent patterns. In this study, we chose $\alpha = 0.3$ as it produced mostly contiguous regions without making spatial constraints dominate the clustering results.





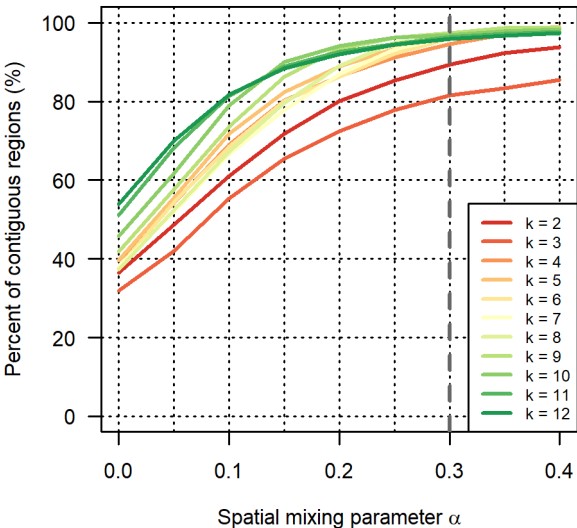

**Figure 3.** The percentage of regions that were spatially contiguous in a grid search over the study period when changing the spatial weight $\alpha$. The optimal value chosen for this study was 0.3 (vertical dashed line).

### 4.2 Fuzziness parameter

The fuzziness parameter $r$ plays a crucial role in balancing the crispness and fuzziness of clusters, ensuring they are neither overly sharp (all membership values are 0 or 1) nor completely fuzzy (all membership values are $1/k$). We used cluster validation metrics from the *fpc* package for R (Hennig, 2023) to evaluate the clustering results from the grid search. Among these metrics, three were particularly informative indicators of cluster quality:

- Within-between ratio: This metric evaluates cluster separation and compactness. The numerator calculates the sum of
squared distances between data points and their respective cluster centroids, indicating compactness. The denominator computes the sum of squared distances between cluster centroids and the centroid of the entire dataset, indicating separation. A lower ratio suggests well-defined clusters with distinct separations.

- Average silhouette width: Silhouette widths assess the coherence of each data point within its cluster by comparing its distance to other points within the same cluster against distances to points in neighbouring clusters. The average
silhouette width measures whether clusters have clear boundaries. Values near 1 indicate appropriate clustering, while values near -1 suggest overlapping clusters with potential misclassifications.

- Normalized gamma: This metric evaluates the quality of a clustering result by examining its ability to organize data points into meaningful clusters based on pairwise distances. A reference distance matrix is produced with binary values





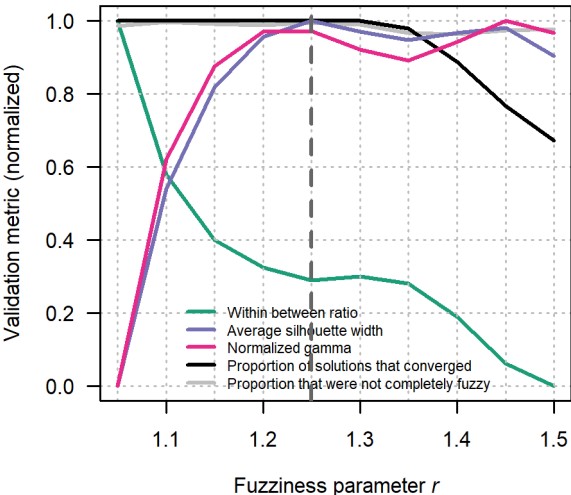

**Figure 4.** The impact of varying the fuzziness parameter $r$ on fuzzy analysis clustering results. Average values of within-between ratio, average silhouette width, and normalized gamma from the grid search are plotted for different values of $r$. Metrics are normalized between their maximum and minimum values to emphasize relative maxima and minima. The proportion of results that converged is denoted by the black line and the proportion of results that were not completely fuzzy is donated by the gray line. The optimal $r$ value chosen for this study was 1.25 (vertical dashed line).

indicating whether pairs of data points belong to the same or different clusters. Normalized gamma is the correlation
between these reference distances and the actual distances between data points (i.e., $dist$). Values close to 1 indicate
agreement between distances and cluster memberships, indicating a higher-quality clustering result.

We calculated the validation metrics for every clustering result generated by the grid search. Additionally, we logged
warning messages from the fanny clustering function to detect instances of non-convergence or when memberships approached
complete fuzziness. These occurrences are direct indicators that the fuzziness parameter was either too small or too large,
respectively.

The average silhouette width and normalized gamma metrics favoured $r$ values of 1.2 and above, indicating improved
clustering coherence and stronger agreement between distances and cluster memberships within this range. The within-between
ratio also showed enhanced performance with increasing $r$, suggesting tighter and more well-defined clusters. An elbow in the
within-between ratio graph suggested similar outcomes for $r$ values between 1.2 and 1.35. Warning messages about complete
fuzziness began at $r = 1.35$ and became more common as $r$ approached 1.5. While convergence errors occurred for all $r$ values,
they were least frequent for $1.05 < r < 1.3$.





To balance crispness and fuzziness, we chose $r = 1.25$ as the optimal value for our dataset, as it offered a meaningful level of uncertainty while promoting well-structured clusters with clear boundaries.

### 4.3 Number of clusters

We considered two approaches for selecting the optimal value of $k$ from the grid search: optimizing clustering validation metrics and aligning the number of clusters with the number of human-assessed regions. The within-between ratio decreased with increasing $k$ (Fig. 5a), while the average silhouette width and normalized gamma reached peaks at intermediate $k$ values (Fig. 5b-c). Plotting average silhouette width or normalized gamma on individual days (not shown) found these metrics had relatively flat peaks, indicating that selecting $k$ from the maximum value of these metrics could result in arbitrary and fluctuating
clustering results on consecutive days.

A more favourable strategy for selecting the optimal number of regions involved choosing the smallest $k$ when a specified metric surpassed a predefined threshold. This aimed for smaller, consistent $k$ values over time. We used the number of human-assessed regions to determine these thresholds by selecting grid search cases from each day when $k$ equalled the number of human-assessed regions. Selecting $k$ when the within-between ratio was below 0.65, the average silhouette width exceeded
0.27, or the normalized gamma exceeded 0.54 would, on average, produce a similar number of regions as human forecasters.

We found selecting $k$ with an ensemble approach of multiple metrics was more effective than using any single metric. This approach identified the $k$ value that met the threshold criteria for each validation metric, then averaged and rounded these $k$ values to determine the optimal number of forecast regions for that day.

### 4.4 Sequential weight

We implemented sequential clustering by introducing a sequential weight $\beta$ that considered the previous day's clustering results. This involved computing distances $dist_{seq}$ from the previous day's clustering membership vectors $u_{iv}$ using the maximum difference between vector components (supremum norm method). A grid search with sequential clustering over the study period was conducted for different $\beta$ values ($\beta = \{0, 0.01, ..., 0.1\}$). We evaluated the results for the 107 days when data was available on consecutive days.

We evaluated performance for each value of $\beta$ by counting the number of times forecast regions changed, and by quantifying the complexity of the changes with the Adjusted Rand Index. The Adjusted Rand Index quantifies the similarity between two clustering results by comparing how data points are grouped (Hennig, 2023). A value of 1 signifies an identical assignment of data points to clusters and -1 indicates completely different clusters. We computed similar metrics for human-assessed forecast regions on the same days, offering a benchmark to gauge changes in region arrangements across different values of $\beta$.

The number of human-assessed forecast regions changed 12 times over 107 days, with region arrangements changing on 34 days. The average Adjusted Rand Index value was 0.94, indicating infrequent and simple changes. In contrast, clustering without sequential clustering ($\beta = 0$) resulted in the number of regions changing on 57 days and arrangements changing on 97 days, with the average Adjusted Rand Index at 0.69, suggesting frequent and complex changes in the regions. Such frequent




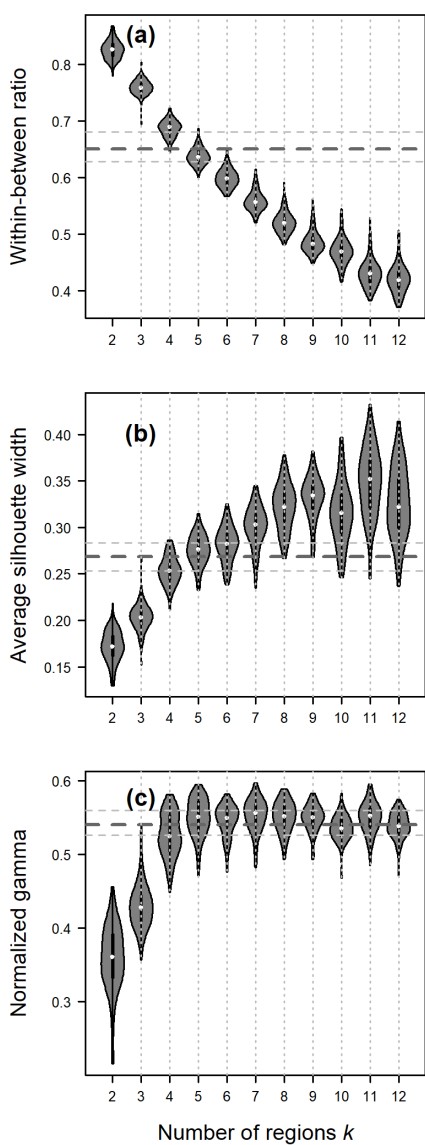

**Figure 5.** Performance of clustering results for different numbers of regions $k = \{2, ..., 12\}$ over the study period based on (a) the within-between ratio, (b) the average silhouette width, and (c) the normalized gamma. Horizontal dashed lines represent each metric's median and interquartile values when $k$ equalled the number of human-assessed regions.



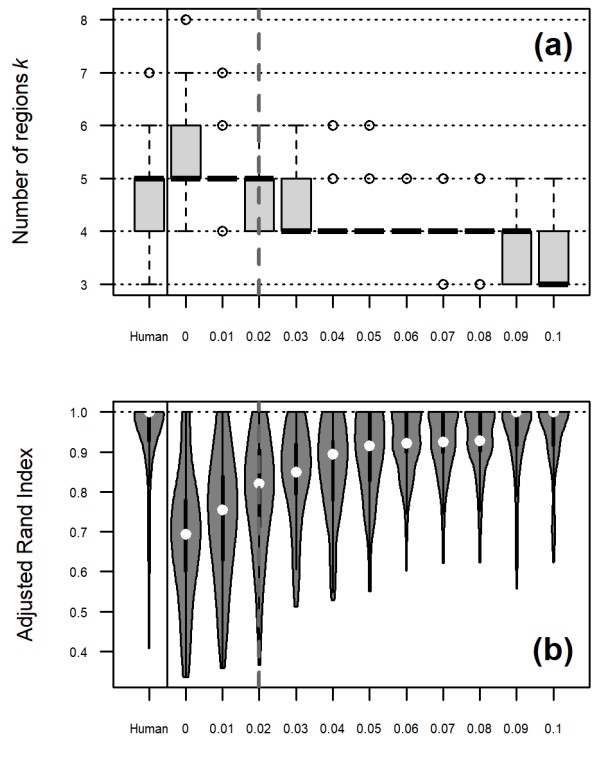

**Figure 6.** Quantifying changes in clustering results on consecutive days: (a) Number of forecast regions over the season, and (b) Adjusted Rand Index measuring the similarity of clustering results on consecutive days. The leftmost plots display the distribution of values for human-assessed forecast regions as a benchmark, followed by non-sequential clustering ($\beta = 0$), and sequential clustering with $\beta$ values ranging from 0.01 to 0.1. The optimal $\beta$ value chosen for this study was 0.02 (vertical dashed line).

rearrangement of regions is impractical for operational forecasting, highlighting the need for sequential clustering to stabilize

the results.

Applying sequential clustering led to fewer and less drastic changes on consecutive days, especially as $\beta$ approached 0.1 (Fig. 6). Large values of $\beta$ tended to decrease the number of regions over the season and forced clustering solutions to converge to a stable solution and remove responsiveness to changing conditions. We selected $\beta = 0.02$ to balance result stability with responsiveness to significant changes in snowpack conditions, recognizing that the optimal value could depend

on the forecasting context. With $\beta = 0.02$, the number of regions changed on 34 days, the arrangements changed on 93 days, and the average Adjusted Rand Index was 0.79, reflecting a moderate complexity of changes compared to human forecasters and non-sequential clustering.





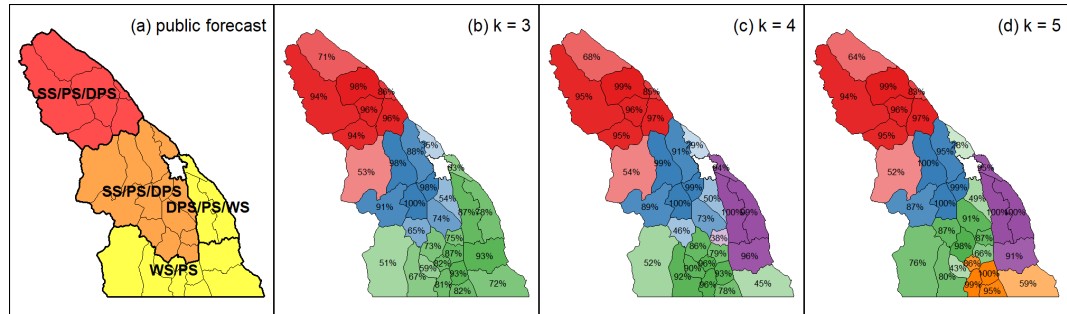

**Figure 7.** Map of (a) human-assessed forecast regions on February 3, 2023 colour-coded by treeline danger rating (red = 4-High, orange = 3-Considerable, yellow = 2-Moderate) and labelled with avalanche problems in order of importance (SS = storm slab, WS = wind slab, PS = persistent slab, DPS = deep persistent slab). Clustering results for (b) $k = 3$, (c) $k = 4$, and (d) $k = 5$ regions are shown with subregions colour-coded by their primary cluster membership with greater transparency for low membership values and their membership values labelled.

## 5   Clustering results

This section demonstrates the clustering method's capability with examples from the 2022-23 winter. Using the optimized
parameters from Section 4, these results serve as a case study rather than a comprehensive cross-validated evaluation.

### 5.1   Clusters for February 3, 2023

The February 3, 2023 clustering results highlight the method's effectiveness in partitioning meaningful forecast regions. On this day, the Columbia Mountains had four human-assessed forecast regions with varying avalanche hazard conditions (Fig. 7a). The northernmost region had a treeline danger rating of 4-High, while the central region was 3-Considerable and regions
in the south and east were 2-Moderate. Avalanche problems varied across regions, with storm slabs posing the primary problem in the regions with High and Considerable danger, while wind slabs and deep persistent slabs were the primary problems in regions with Moderate danger. Persistent slabs were the secondary problem in all regions, with deep persistent slabs also listed as a third problem in the northern and central regions.

The results for $k = \{3, 4, 5\}$ demonstrate the clustering method's ability to partition regional patterns at different resolutions
(Fig. 7b-d). These regions generally correspond to major avalanche hazard patterns assessed by forecasters. For $k = 3$, regions align with danger rating trends, while $k = 4$ and $k = 5$ further divide areas with Moderate danger, potentially reflecting distinct snowpack conditions and avalanche problems in these areas. Fuzzy cluster memberships are most pronounced near region borders, with some subregions shifting their primary membership as $k$ changes, particularly in southern areas. The maps of memberships for each cluster region further illustrate how fuzzy analysis clustering can reveal overlapping patterns, as some
subregions exhibit similar membership to multiple regions (Fig. 8).





**Figure 8.** Each region produced with $k = 4$ clustering on February 3, 2023 is shown with a map of the memberships of each subregion to that region, an average snow profile from all subregions with membership values above 75 %, a textual summary of snow depth, 3-day snowfall, and unstable persistent weak layers (average values are provided first followed by the minimum and maximum values in brackets), and finally, the grain type profiles for all subregions that have the strongest memebership with that region.





The snow profile characteristics for the $k = 4$ clustering results illustrate the primary factors driving the partitions (Fig. 8). Similar plots for $k = 3$ and $k = 5$ are provided in Appendix B. Distinct snow depth patterns are clear, with deep snowpack areas separated from shallow ones. The northern region (Region 1) had the greatest amount of 3-day snowfall (12 to 25 cm), compared to the central region (Region 2) with 4 to 18 cm, and the other regions with less than 8 cm. Great amounts of 3-day 245 snowfall in the northern and central regions align with their elevated danger ratings and storm slab problems.

All subregions contain faceted grains or depth hoar near the bottom of the snowpack (Fig. 8), which aligns with the deep persistent slab problem listed in all regions except the southernmost region. Forecasters did not assess a deep persistent slab problem in the southern region on February 3 because melt-freeze crusts in the upper snowpack reduced the likelihood of triggering. These crusts are present in the simulated profiles. In the eastern region (Region 4), 4 of 5 profiles had unstable 250 persistent weak layers, while the other regions had smaller proportions of unstable persistent weak layers (Region 1: 4 of 8; Region 2: 4 of 9; Region 3: 2 of 10). These proportions align with the fact that deep persistent and persistent slab problems were the most important problems in the eastern region but were secondary problems in other regions.

## 5.2 Temporal patterns

Sequential clustering over the season resulted in gradual changes in the number and arrangement of forecast regions (Fig. 9). 255 Some subregions formed consistent groupings with high membership values over the season, especially in the northern and central areas. In contrast, the southern and eastern areas were more variable with subregions showing sustained low membership values that caused them to fluctuate between regions.

## 5.3 Comparison with human forecast regions

To compare the clustering method's typical forecast regions with public forecasts (Fig. 10), we identified common arrangements 260 by counting how often each pair of subregions was grouped together. Using these counts, we applied the fanny clustering method with $k = 4$ and the default fuzziness parameter $r = 2$ to generate groups representing the four most frequent forecast region arrangements over the study period.

The clustering method consistently grouped subregions into similar regions as human forecasters (Fig. 10). These regions' arrangements roughly match the patterns observed on February 3, 2023, as the conditions that day were representative for 265 most of the 2022-23 season. However, for some specific subregions there were differences between the clustering and human forecast regions. Discussions with Avalanche Canada forecasters revealed two main reasons for these differences. First, some of these subregions have limited data availability, leading to lower confidence in forecasters' assessments. Second, some were areas where the operational snowpack model had known accuracy issues, such as underestimating snowfall. Either case could cause inaccurate arrangements, and it is not clear which solution would better align with reality.





**Figure 9.** Clustering results for each day between December 1, 2022 and February 10, 2023. Subregions within the clusters are colour-coded based on their primary cluster membership, with lower membership values indicated by greater transparency. Human-assessed forecast regions are outlined in black.



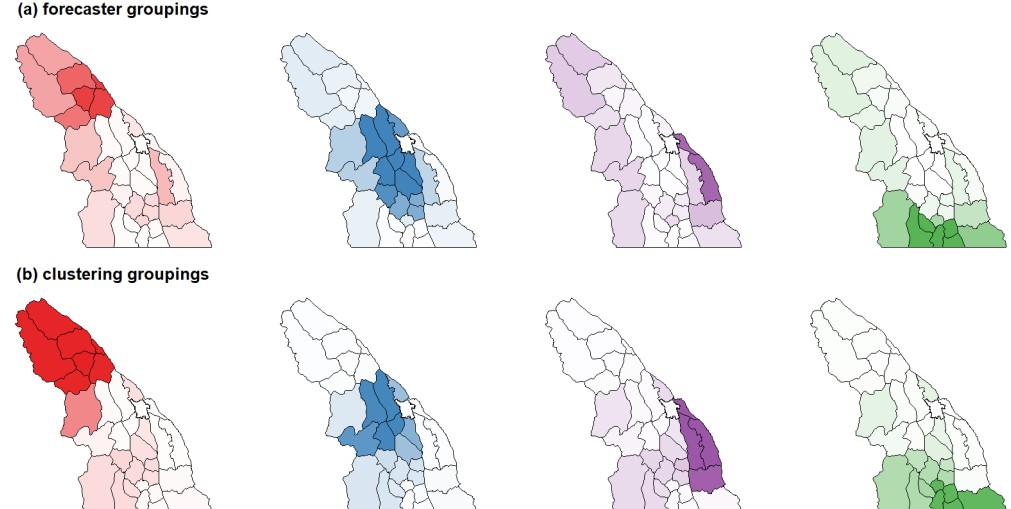

**Figure 10.** The four most common arrangement of subregions for the 2022-23 season according to (a) human forecasters and (b) clustering results.

## 6 Discussion

### 6.1 Quality of clustering results

Clustering simulated snow profiles effectively captured major hazard patterns in the Columbia Mountains during the 2022-23 winter season. The clustering of subregions into forecast regions closely aligned with human-assessed regions (Fig. 10). On February 3, 2023, these groupings captured differences in avalanche danger ratings and avalanche problems across the Columbia Mountains (Fig. 7 and 8). The fuzzy analysis clustering method conveyed the inherent uncertainty associated with simulated snow profiles, making it more suitable than deterministic clustering methods. Clustering over the season suggested that the number and arrangement of forecast regions could change more often than the human-assessed region arrangements.

A limitation of this study was the clustering results were presented with the same dataset used for parameter optimization. While cross-validation across multiple seasons would provide a more rigorous evaluation, our goal was to demonstrate the potential value of this method for avalanche forecasting through a case study.

### 6.2 Practical avalanche forecasting considerations

Clustering could help forecasters identify spatial patterns in complex datasets such as snowpack model simulations. While a similar approach could be applied to traditional field observations, spatially distributed snowpack simulations provide the advantage of continuous spatial and temporal coverage.

The operational snowpack model used in this study was primarily configured to predict avalanche problems associated with new snow and persistent weak layers. Consequently, the snow profile distance metric $dist_{pro}$ emphasized these specific snow




profile characteristics. However, this distance metric could be changed to incorporate other relevant characteristics, such as those associated with wind slab or wet snow problems. Furthermore, expanding this distance metric to also integrate field observations could provide a more comprehensive understanding of hazard patterns.

The clustering results presented here focus on regional-scale patterns, as the rows and columns in the distance matrix represent entire subregions. However, the concept of spatial constraints can be extended to other spatial scales by adapting the distance metric $dist_{geo}$ to quantify other types of spatial relationships. For example, $dist_{geo}$ could be redefined to measure the distance between profiles on different aspects and elevation bands, or between profiles distributed across a single slope. Integrating aspect and elevation bands into the clustering analysis would enable a more comprehensive representation of the

spatial scales relevant to forecasters. For example, Bouchayer (2017) demonstrated that clustering simulated snow profiles on a 1.3 km grid in France revealed local-scale snowpack patterns and elevation effects, highlighting the potential of incorporating more spatial considerations into clustering analyses.

While clustering offers insights into complex model output, forecasters should treat them with some level of caution. Due to the challenge of validating the accuracy of spatially distributed snowpack simulations, we currently do not recommend

using this clustering method for unsupervised automation. Instead, forecasters should consider clustering as a data exploration tool. For example, forecasters could adjust the number of regions $k$ to view clustering results at different resolutions and gain insights into potential hazard patterns without blindly relying on automated processes.

### 6.3 Technical considerations for snow profile clustering

A critical aspect of this clustering method was the distance metric used to compare snow profile characteristics, which took

advantage of the recent developments of Herla et al. (2022) and Mayer et al. (2022). Condensing snow profile comparisons into a single numerical value is inherently challenging and represents a serious simplification. Hence, careful consideration must be given to quantifying snow profile distance, given the impact it can have on clustering results. Deriving a meaningful snow profile similarity metric for this study required meticulous trial-and-error to properly weigh relevant snowpack features.

The distance between subregions $dist$ can easily integrate into other clustering methods such as hierarchical clustering or

partition-based methods like k-means and k-medoids. Hierarchical clustering generates intuitive tree-like structures with nested clusters, visualizing patterns at different resolutions. Herla et al. (2021) presented a simple example of hierarchical clustering of snow profiles. An enhancement to k-means clustering could involve applying dynamic barycenter averaging to define cluster centroids (Petitjean et al., 2011), as Herla et al. (2022) recently adapted this method for snow profiles. Additionally, clustering simple scalar indices derived from snow profiles would be more computationally efficient than evaluating the entire snow

stratigraphy. For example, Reuter et al. (2023) derived avalanche problem types from simulated snow profiles and clustered their frequencies to predict snow climatologies.

Selecting parameters for a clustering method must be done with care for each application. Sect. 4 presents possible approaches for tuning parameters to test data. Factors such as the variability within a snow profile dataset, the number of subregions, and their spatial arrangement will influence parameter selection. Recent attempts to apply this method across western Canada



suggest that the parameters may need re-tuning to accommodate other datasets. Tuning parameters to make the clustering results align with human-derived forecast regions proved to be helpful.

Computational time is a critical consideration for operationalizing clustering methods. While computing pairwise similarities for a small number of profiles is relatively efficient, scalability becomes an issue with larger datasets. Applying different clustering methods or changing $k$ is relatively fast after computing the distance matrix. Real-time applications should consider
code optimization and parallelization to manage computational demands efficiently.

## 7    Conclusions

Statistical clustering offers a valuable approach for identifying avalanche hazard patterns within complex snowpack model datasets. This study shows the effectiveness of a clustering method that accounts for spatial and temporal trends, as the major patterns across the Columbia Mountains during the 2022-23 winter season aligned with human-assessed forecast regions.
The application of fuzzy analysis clustering facilitates the representation of uncertainty in simulated snow profiles, providing nuanced insights for forecasters. Adjusting the number of clusters can reveal patterns at various levels of spatial resolution.

These methods can adapt to consider different criteria, such as different snowpack characteristics or spatial relationships. With numerical snowpack modelling advancing rapidly, forecasters need intuitive tools to explore model outputs. Avalanche Canada plans to implement and refine these methods in their operational snowpack model system. Embracing clustering as a
form of exploratory data analysis should enhance the interpretability of snowpack model outputs and support more informed decision-making in avalanche forecasting.

**Footnotes**

– **Code and data availability:** Code and data are publicly available on the Open Science Framework at https://osf.io/4u2az (Horton et al., 2024).
– **Author Contribution:** All authors conceptualized the research with SH leading the analysis and writing, FH developing many underlying methods, and PH providing supervision and proofreading.

– **Competing Interests:** The authors declare that they have no conflict of interest.

– **Acknowledgements:** We thank Avalanche Canada for providing operational forecast data and feedback on the clustering results and Patrick Mair for guidance on clustering methods.
– **Financial Support:** NSERC, Mitacs, and Avalanche Canada funded this research.

## Appendix A:  Configuration of operational modelling system

This appendix summarizes Avalanche Canada's operational snowpack modelling system for the 2022-23 winter (Horton et al., 2023). The system forced SNOWPACK version 3.4.5 (Lehning et al., 1999) with meteorological data from two numeric weather prediction (NWP) models run by Environment and Climate Change Canada: The High-Resolution Deterministic Prediction



System (2.5 km horizontal resolution) and the Regional Deterministic Prediction System (15 km resolution) (Milbrandt et al., 2016).

To capture regional-scale patterns across large forecast regions, the system selected representative grid points from each NWP model with a stratified sampling approach. Mountainous areas were divided into small microregions, from which up to three grid points were sampled to represent weather conditions at alpine, treeline, and below treeline elevations. This study

used 168 treeline elevation grid points, including 126 from the high-resolution NWP model and 42 from the regional model (see Fig. 1a).

Hourly time series data for precipitation amount, precipitation type, air temperature, humidity, wind speed, incoming shortwave radiation, and incoming longwave radiation were compiled six hours at a time from each successive NWP model run to generate the necessary meteorological forcings for SNOWPACK. SNOWPACK was configured to simulate flat field snow

profiles with wind transport disabled, ensuring simulations represented widespread regional snowpack characteristics.

As part of the operational model, snow depth observations were assimilated weekly following a method based on Horton and Haegeli (2022). The method compares changes in modelled snow depth over the previous week with changes observed at nearby sites (i.e., automated weather stations and manual observations by avalanche professionals). Snow depth observations from these sites were lapse rate adjusted to local treeline elevations and then spatially interpolated to the model grid points.

Potential errors in snowfall amounts were identified by comparing modelled snow depth increases over the past week with increases in interpolated observations. Cases where either the modelled or observed snow depth increased by more than 10 cm were identified, and then corrective action was taken if the increases differed by more than 10 %. Specifically, SNOWPACK was rerun with the input precipitation amount adjusted by a constant factor to nudge the modelled snow depth towards observed values.

Simulated snow profiles were stored in a database, which fed an online visualization dashboard used by operational avalanche forecasters. For this study, we queried a subset of profiles from this database.

**Appendix B: Clustering results on February 3, 2024 for other values of $k$**

.





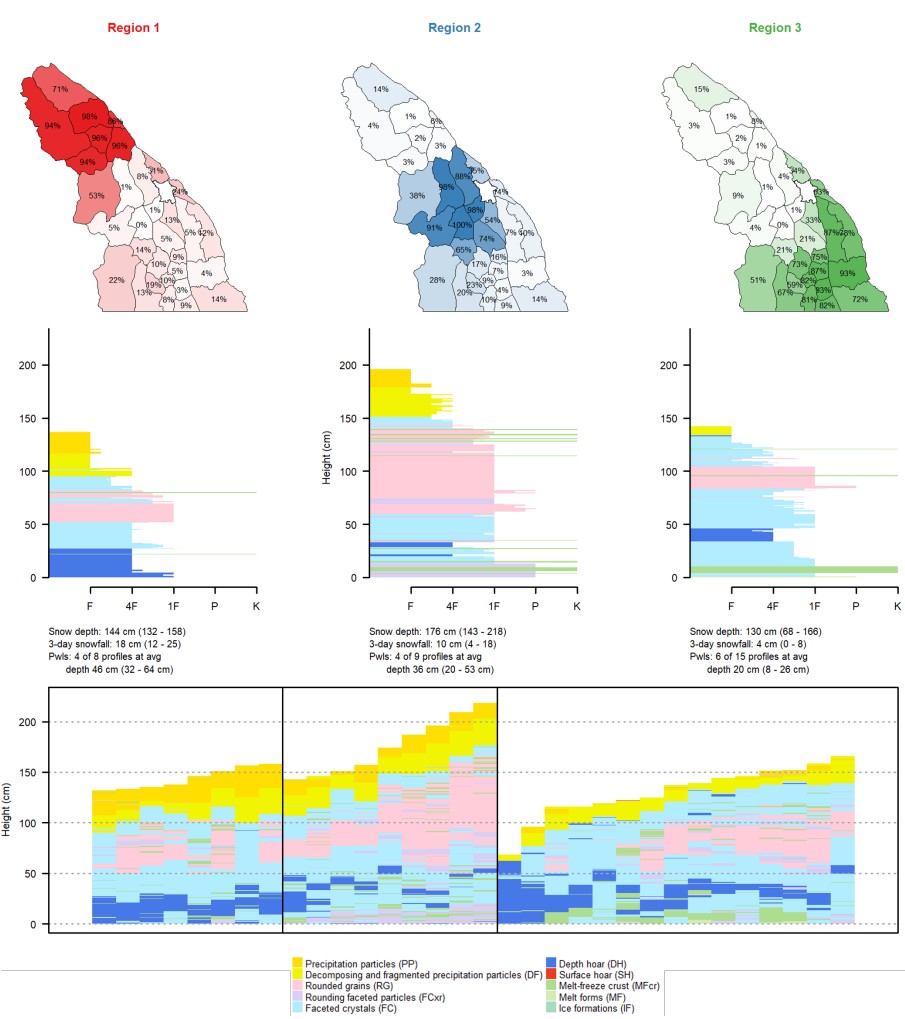

**Figure B1.** Comparison of regions produced with clustering results for $k = 3$ on February 3, 2023. Each column summarizes a region with a map of the memberships of each subregion to that region, an average snow profile from all subregions with membership values above 75 %, a textual summary of snow depth, 3-day snowfall, and unstable persistent weak layers (average values are provided first followed by the minimum and maximum values in brackets), and finally, the grain type profiles for all subregions belonging to that region.





**Figure B2.** Comparison of regions produced with clustering results for $k = 5$ on February 3, 2023. Each column summarizes a region with a map of the memberships of each subregion to that region, an average snow profile from all subregions with membership values above 75 %, a textual summary of snow depth, 3-day snowfall, and unstable persistent weak layers (average values are provided first followed by the minimum and maximum values in brackets), and finally, the grain type profiles for all subregions belonging to that region.



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
