# Peer review of "Clustering simulated snow profiles to form avalanche forecast regions"

_EGUsphere, 2024_

## Author Response (AR1)

**Author's response to "Clustering simulated snow profiles to form avalanche forecast regions"**

**CC1**

Our revised paper addresses their three questions by citing the area of the Columbia Mountains study area (Sect 2.1), clarifying the number of grid points (Appendix A, Fig. 1a), and discussing the potential impact of our operational clustering product influencing forecaster assessments (Sect. 6.1).

**RC1**

We have addressed the reviewer's comments in the revised manuscript. Key changes include (1) simplifying the parameter selection method and (2) presenting additional results from the 2023-24 season. These changes make the method more broadly applicable by reducing the complexity of parameter tuning and demonstrating its performance on independent data. We also added a few sentences in the Discussion clarifying limitations related to parameter tuning.

Here's how we addressed each specific comment:

- *Figure 4 is not clear in black and white. Whereas being readable in black & white is too much to ask for figures 7-10, for this figure a simple linestyle change (dotted/dashed or markers) may make it readable in black and white (for example on an eReader.)*

We changed Fig. 4 to black and white.

- *Is there a reason why Beta values > 0.1 were not investigated? If so please mention/explain.*

We expand the explanation of beta values in Sect. 4.4 and extended the grid search to include values up to 0.25. These higher values clearly show the issue of rapid convergence to fixed regions that do not change over time.

- *Line 210: "The number of human-assessed forecast regions changed 12 times over 107 days, with region arrangements changing on 34 days." The into mentions "115 days when both model and human data were available for analysis." What is this discrepancy (107 vs 115)?*

We clarified that sequential clustering was only applied on days with data available on consecutive days, as opposed to non-sequential clustering which can be applied to every day.

- *Fig 6: It is not clear to me how the ARI supports the choice of Beta=0.02. I see how 6a supports that if the goal is to mimic the human assessment of the nr of regions. But 6b is unclear (to me). Intuitively I would say the ARI should be high on days where nothing*

*changes, but you want the clusters to change only when the snowpack changes. How does ARI reflect that?*

We simplified this grid search by holding the number of regions fixed at k = 5 to remove variability from the changing number of regions. Fig. 6 was simplified to show (a) the number of times the region arrangements changed and (b) the distribution of ARI values over the season. Ultimately the choice of optimal beta is still subjective, but these plots better illustrate that 0.02 produces a midway complexity between human and non-sequential clustering.

- *Sect 5.3: "default fuzziness parameter r=2": previously (sect 4.2) r was determined/picked to be optimal at 1.25; why is it 2 here?*

We add a sentence to explain why a larger fuzziness parameter was needed for the larger number of zeros in count data as opposed to the inherently fuzzier patterns in snow profile data.

- *Out of curiosity: did you see the clustering change as solar irradiation (and thus the difference between N and S) became larger, i.e. throughout the season?*
- *In general; how do you make representative profiles for a region when solar irradiation leads to big differences in aspect? Or is the clustering only done for dry (ENW) avalanche profiles?*

See our previous author comment (AC2) for an in-depth discussion on solar radiation and aspects. The limitation of our analysis focusing on flat, sheltered profiles is discussed in both the Data (2.2) and Discussion (6.2) sections.

**RC2**

We have addressed the reviewer's comments in the revised manuscript. Key changes include (1) simplifying the parameter selection method and (2) presenting additional results from the 2023-24 season. These changes make the method more broadly applicable by reducing the complexity of parameter tuning and demonstrating its performance on independent data. Additions to Sect 6.1 acknowledge this study is not a comprehensive cross-validation.

Here's how we addressed each specific comment:

- *Line 59 you limit the study to flat, sheltered terrain at treeline elevation. This seems an interesting selection here, but therefore the results are compared to human assessed forecast (e.g. Fig 7a). This human assessed forecast should take into account all elevations. Thus, is this really comparable ? In the discussion, you underline the high impact on elevation on the results of clustering (Bouchayer, 2017, line 295 and following).*

Flat, sheltered treeline profiles capture most processes driving new snow and most persistent weak layer type problems in the Columbia Mountains. Wind-drifted snow and wet snow problems are less likely resolved, but the former two tend to provide a good measure of danger level and region arrangements for two reasons (1) they have a larger influence on danger level, (2) their spatial distribution tends to be more widespread across elevations and aspects on a regional scale. We expanded this explanation to our Data and Discussion section, including the idea of extrending the

snowpack distance metric to account for wind drift and extending the spatial distance metric to account for patterns across aspects.

- *Line 60, you state that the method used to obtain simulated profiles is of little relevance. I do not agree. I understand the idea to put it in appendix not to break the flow of the paper (which is quite easy to read). However, choices in the simulated profiles can highly impact the results. Maybe this sentence could be modified. In particular, I noted line 355 that you selected meteorological data from 2 models. I wonder if this could impact the results as the climatology of meteorological data may be different between models, especially as the resolution are very different (so the representation of mountainous area is significantly different). I would be at least interested to have a map with the selected points for each model to ensure that there is no obvious correlation between clustering results and the choice of meteorological model.*

We removed the "limited relevance" part of the sentence and updated the plot symbols in Fig. 1a to show profile locations for each NWP model. The coarser 10 km model has fewer points reaching treeline elevation, and tends to have more data points in the eastern subregions where the terrain is higher, compared to the 2.5 km model which has treeline elevation points across the study area. For instance, the 10 km model only had points in 16 of 32 subregions. We did not see significant impacts from the different models since each subregion's representative profile was the average of all treeline profiles and the high-resolution model profiles tended to dominate the average due to their higher number in most subregions (except one). We add a mention of this averaging effect in Appendix A.

- *Line 100-103: Could you explain why you state that "this method closely aligns with forecasters' criteria..."?*

We revised our wording in this paragraph. This selected similarity method is better aligned with "avalanche forecasting" by incorporating both mechanical properties, such as layer instability, and structural properties, like grain type and hardness. Unlike other methods that focus mainly on structural properties and may overemphasize thick cohesive layers, this approach appropriately weights thin but unstable layers more heavily.

- *Line 109-110 "that are more likely to align... than would result from basic Euclidian distances": a distance between two areas may be defined differently than the distance between centroid, especially it could be defined as the minimum distance between two points, each on in each area. This latter Euclidian distance would be relevant. Maybe this part of sentence is not relevant here, you made a choice that seem simple but relevant for your goal.*

Interesting point about computing Euclidean distances from polygon edges rather than centroids. We added a sentence to be more specific about our tests with Euclidean distances and the advantage of the neighbourhood distance for matching snow climates.

- *Equation 2: Notation $u^r_{iv}$ may not be clear. I supposed this is $u_{iv}$ at power r. If so maybe, the power could be put more clearly with a parenthesis, for instance.*

Parenthesis added.

- *Line 130: What is the value of threshold tolerance? Do you use a usually used value or how do you select this value?*

We add a sentence to state we use the default threshold value in R 10^(-15).

- *Line 143: How do you select these ranges.? Especially the range for r is surprising as you said later that the default fussiness parameter is 2 (line 261).*

We added the statement, "The rationale for these values is explained in the following sections," and provided detailed explanations for the parameter ranges in their respective subsections. We also expanded the grid search range to better illustrate the issues that arise when exceeding the parameter limits in our initial manuscript.

- *Line 159: you here use distance between centroid whereas you stated line 109-110 that it was not relevant*
- *Line 163: could you provide a reference and/or explain the goal of this metric. I am not sure I understand.*
- *Same as 9 for line 167-171. "high-quality" clustering is not sufficient for me to understand the specificity of the score.*
- *Line 179: "An elbow in the within between ratio... between 1.2 and 1.35": The within between ratio decrease after 1.35 and lower is better for this ratio, so I would not have chosen values between 1.2 and 1.35 with this ratio.*

Our revised parameter selection only uses "average silhouette width", so comments about other metrics are no longer relevant. We expand our explanation of this metric in Sect. 4.2 and cite Kaufman and Rousseeuw (2009).

- *Line 196: Determination of k value. Why do you select different k values based on different scores and then average the different k values? With such method, you do not have any constraint on the values of the metrics (especially as these metrics are not always monotonous) while using the lower k value that are above/below thresholds for different scores for instance would give stronger results.*

Our revised parameter selection only uses "average silhouette width", allowing for a clean simple explanation of how the threshold criteria can be applied to select k.

- *Line 210-213: "XX days" (used 3 times) is not fully clear. Is this "XX times over 107 days" ? Then, use "times" rather than "days" to be coherent with first use.*

We now report these statistics as a "percentage of days" rather than counts.

- *The average adjusted range index is not defined. Is this the average of adjusted range index for all couples of successive days.*

We revised the previous paragraph to clarify that ARI was calculated for each pair of consecutive days. We now report the median ARI over the season to be consistent with the violin plots, and add "over the season" to make it clear it is the median values over all the ARI values in the sample.

- *Line 213: I suggest "0.69 suggesting \*more\* frequent and complex changes" unless you have a justified threshold on average adjusted rand index to separate between complex and simple clustering.*

Change made.

- *Figure 6: You here have both the influence of k and beta on the adjusted rand index (ARI). If the number of cluster is higher in the automatic clustering compared to reference human clustering but lower adjusted rand index even though the clustering is perfect because some human areas are partitioned into several clusters that varies against days while the union of clusters do not vary and match the human clusters. Why not doing as for other parameters and separate the effect of k and beta by fixing k to have a relevant comparison of ARI values?*

We adopted this suggestion and held k fixed at k = 5 to isolate the effect of beta. Fig. 6a now counts the percentage of days regions change arrangement rather than count the number of times k changes.

- *Figure 7 and 8 top: please report the limits of public forecast on clustering maps to ease the reading of the graph.*

We added thick black lines outlining human-assessed regions to all relevant maps.

- *Figure 7 and 8 top: You do not discuss the differences between human clustering and automatic clustering. For some misclassified regions, the fuzzy clustering show the uncertainty (e.g. 54% in the south of northern region) however, in some cases, there is no uncertainty shown (e/g/ >90% for south of eastern cluster and some in the south of the center region). Taking human clustering as the reference may be one limit here that could be discussed or maybe it is a limit of the method. Such result make me want to know more! However, I understand that such details may not be the core of such paper and should not take too much space.*

The thick black lines around the human-assessed regions aid visual comparison in these figures. In Sect. 5.1, we added a sentence to acknowledge the presence of some high-membership subregions that differed from the human-assessed regions. In Sect. 5.4, we discuss these discrepancies in general terms and suggest possible causes for the differences.

- *Line 288-289: "Furthermore, expanding this distance...": I do not understand this sentence and what you have in mind. Maybe could you develop?*

We removed this sentence. This referred to another complex idea that is outside the scope of this paper.